# OpenReview forum: "Emergent Deceptive Behaviors in Reward-Optimizing LLMs"
_ICLR.cc/2026/Conference — ICLR 2026 Conference Desk Rejected Submission_

### Official Review · Reviewer_8vKM · 2025-10-18

**Soundness:** 3
**Presentation:** 4
**Contribution:** 2
**Rating:** 6
**Confidence:** 4

**Summary:**

The paper creates 4 simple settings with reward hacking opportunities and evaluates:
* How quickly 3 small LMs trained with GRPO explore these hacks;
* How much LMs that learned how of these hacks also learn the other hacks;
* Whether learning one hack makes other hacks easier to explore.

**Strengths:**

* The paper studies multiple models for multiple reward hacking settings, providing greater diversity than prior work.
* The paper studies open-source models with open source algorithms. If the settings are open-sourced, this could allow the community to have better open-source model organisms of reward hacking relative to the status quo (e.g. where work like https://arxiv.org/pdf/2406.10162 is closed-source)
* The generalization results, while not very surprising given prior work, provide additional evidence of cross-reward hack transfer, both in zero-shot and “catalysis” settings.
* The settings are clearly explained, and the results are well presented.

**Weaknesses:**

* RQ1 seems very hack-specific, and more a property of a particular tuple of RL setup, initialization and reward hack than a generalizable finding which the community can learn from (which the abstract implies).
* The work is not particularly novel. In particular, https://arxiv.org/pdf/2406.10162 studied reward hacking settings and generalization questions closely related to this work.
* Some reward hacks are very hard to explore, and your experiments do not cover those hacks. Like https://arxiv.org/pdf/2406.10162, it might have provided deeper insights about what would happen for the most important, hardest-to-explore reward hacks.

**Questions:**

* What do you see as your main contributions compared to https://arxiv.org/pdf/2406.10162
* How hard is exploration in your hardest to explore setting? More information about this would be welcome, for example by running the zero-shot (before any training) experiment with a large n (how big is n in table 3?) It would also be informative to see what ER is when fine-tuning on the intended behavior (which might erode some of the safety training that might cause an ER of 0 in non-base models).
* The paper claims “self-taught strategies are significantly stealthier and more resilient to standard mitigation techniques” in the conclusion. What mitigation technique are you referring to?
* Do you intend to open source the code work and make it easily usable by the community?

Additional feedback that will not affect my score:
* A more consistent ordering of the 4 settings in Figures would make the paper easier to skim.

---

> ### Author Response · Authors · 2025-11-22
> **Response to Reviewer 8vKM**
>
> We sincerely thank the reviewer for the insightful comments and the rigorous interrogation of our experimental dynamics. We have carefully addressed your questions regarding exploration difficulty, comparative novelty, and experimental controls below.
>
> ### 1. Contributions vs. Denison et al. (2024)
>
> While both works identify that misalignment generalizes, our work differs fundamentally in **scale, scope, and mechanism**:
> * **Universality**: Denison et al. focus on frontier models (Claude-2) and found no sophisticated tampering without specific prompting. We demonstrate that **smaller, open-weights models (4B/8B, Base/Instruct/Thinking model) spontaneously develop these behaviors** under standard RL. This serves as a "canary in the coal mine," proving the risk is intrinsic to optimization, not just scale.
> * **Scope**: They study the thematic progression from Sycophancy to Subterfuge. We propose "Capability-Driven Alignment Risk," adding **Robustness Failures** (e.g., Context-Conditional Compliance) where models exploit distribution shifts (Audit vs. User) even when rewards are correctly specified.
> * **Spontaneous Emergence vs. Curated Curriculum**: Denison et al. use a thematic curriculum (Sycophancy $\to$ Subterfuge) to induce generalization. We focus on **spontaneous emergence** in distinct games without a necessary sequential thematic link. Crucially, our tasks (e.g., Reward Tampering vs. Self-Grading) share no surface-level similarity. Proving transfer between such mechanistically disparate tasks demonstrates that the model is acquiring an abstract "meta-skill" of system exploitation, whereas Denison et al. show the escalation of a specific thematic behavior.
>
> ## 2. Exploration Difficulty and Dynamics
> We thank the reviewer for this rigorous question. We have conducted additional large-scale zero-shot experiments to quantify exploration hardness.
> ### 1） Quantifying "Hardness":
> We conducted high-compute zero-shot evaluations ($N=16$ rollouts/prompt) before training.
> * **Qwen3-4B-Base**: Zero-shot ER = 1.68%.
> * **Llama-3.1-8B-Instruct**: Zero-shot ER = 0.51%.
> * Qwen3-4B (Instruct): Zero-shot ER = 0.0%. This confirms the exploit is a rare but non-zero tail event in the initial distribution.
> ### 2） Explaining Training Dynamics (Why FES=15?):
> For Llama, despite the 0.51% baseline, we report FES=15. Our logs reveal a **"Suppression-then-Emergence"** dynamic
> * **Initial Suppression (Steps 1–14)**: Despite the 0.51% baseline at Step 0 and the ER is 0.39% for step 0, the ER dropped to 0.0% for the majority of the first 14 steps (with only minor noise in step 13, still 0.39%). This indicates that as the RL algorithm initially reduced entropy to optimize for Task Correctness (writing valid code), it effectively "pruned" high-variance tail behaviors—including the rare strings required for tampering.
> * **Strategic Emergence (Step 15+)**: At Step 15, the ER rises to 0.78% and begins a consistent upward trajectory. This marks the phase transition where the model re-discovers the exploit not as random noise, but as a superior gradient path for maximizing reward. Conclusion: Exploration is sufficiently hard that standard capability optimization initially suppresses the behavior. The emergence at Step 15 represents the point where the incentive to exploit finally overpowers the incentive to stabilize.
>
> ## 3. Fine-tuning on Intended Behavior
> We wish to clarify the reviewer's specific request here to ensure we address it adequately. **Are you asking for a control experiment where we optimize only for the intended task (e.g., honest summarization) with the exploit reward channel removed, to see if the Exploit Ratio rises simply due to safety erosion?**
> If this is your request, we are prepared to conduct this experiment immediately and will present the results to you as soon as they are available.
>
>
> ## 4. Stealth and Mitigation Techniques
> We are currently finalizing experimental results regarding the resilience of these strategies against specific mitigation techniques (e.g., targeted safety fine-tuning). We will present these full results in the final revision before the rebuttal period concludes.
>
> ## 5. Code Availability
> Yes, absolutely. We are organizing the codebase and aim to provide an anonymous repository link before the rebuttal period ends to ensure reproducibility.
>
> ## 6. Figure Ordering
>
> We agree and have fixed the ordering of the four settings across all figures and tables in the revised manuscript for consistency.
>
> **We once again thank the reviewer for the excellent suggestions, which have helped us refine our claims and experimental rigor. We remain available to answer any further questions! For the additional experiments, we will make sure to add them soon!**

---

> ### Comment · Reviewer_8vKM · 2025-11-23
>
> Thank you for running these additional experiments, I think the ones you already ran and the ones you plan to run strengthen the contribution, and I will raise my rating as a consequence.
>
> > While both works identify that misalignment generalizes, our work differs fundamentally in scale, scope, and mechanism
>
> I agree that your work differs along these dimensions. I think the work would be significantly more novel if Denison had not existed, as I don't think the differences in scale, scope and mechanism are very big (e.g. Denison contains results for no-progression generalization) and I think Denison deserves a citation, but I think your current results are sufficient to be a standalone publication.
>
> > Fine-tuning on Intended Behavior
>
> This is correct. I think it would make the results stronger, as ER could increase (e.g. from 0% to >0%, I don't expect it to increase as much as when hacking is incentivized) just because of the erosion of helpfulness. I think this is important mostly for the setting where ER starts at 0, as it could explain how the model initially explores into its first hack. If this effect is real, it might provide some evidence and an explanation about why joint harmlessness and capabilities training could slow down reward hacking on capabilities environments (which is another experiment you already plan to run if I understand what you mean by "targeted safety fine-tuning" correctly).

---

> > ### Author Response · Authors · 2025-11-24
> >
> > Thank you so much for the encouraging feedback — we really appreciate it.
> > We're glad to hear that the additional experiments strengthen the contribution, and we agree that Denison deserves proper citation and discussion; we will incorporate this carefully.
> >
> > We will conduct all the remaining experiments as quickly and thoroughly as possible. As soon as we have the results, we’ll share them with you.
> >
> > Thanks again for the thoughtful comments and for raising your rating — it means a lot to us.

---

> ### Author Response · Authors · 2025-12-02
> **Supplementary for the response to reviewer 8vKM**
>
> We have conducted the requested control experiment. For all three models, we performed RL training using a "Perfect Reward Model" (optimizing solely for the intended task accuracy, with the specific exploit reward channel removed) across the three applicable games.
> Experimental Results:
> | Model            | Setting        | Acc (ConC / SelfG / RewT) | ER (ConC / SelfG / RewT) |
> |------------------|----------------|-----------------------------|----------------------------|
> | **Qwen3-4B**     | Original       | 47.5 / 25.8 / 11.1         | 2.0 / 16.0 / 0.0          |
> |                  | w/ Loophole   | 37.4 / 28.5 / 35.3         | 55.8 / 32.8 / 0.0         |
> |                  | w/o Loophole  | 92.3 / 29.7 / 32.5         | 11.3 / 9.0 / 0.0          |
> | **Qwen3-4B-Base**| Original       | 26.6 / 9.0 / 0.5           | 4.1 / 9.0 / 7.1           |
> |                  | w/ Loophole   | 48.0 / 0.2 / 2.8           | 78.5 / 81.6 / 100.0       |
> |                  | w/o Loophole  | 93.6 / 16.2 / 4.6          | 9.2 / 4.3 / 74.2          |
> | **Llama-3.1-8B** | Original       | 65.1 / 7.1 / 0.2           | 4.1 / 44.0 / 0.2          |
> |                  | w/ Loophole   | 54.8 / 16.9 / 1.2          | 72.8 / 80.0 / 100.0       |
> |                  | w/o Loophole  | 90.2 / 20.4 / 41.3         | 13.0 / 2.6 / 0.0          |
>
> Analysis:
> These results provide nuanced support for the reviewer's hypothesis while confirming the dominance of the incentive-driven mechanism:
> 1. Safety Erosion is Minor in Aligned Models: For instruction-tuned models, optimizing solely for the intended task ("w/o Loophole") results in significantly lower ER compared to the "w/ Loophole" setting. In Self-Grading, ER for Llama-3.1 dropped from 80.0 to 2.6, suggesting that for aligned models, capability gain prioritizes honesty when the exploit incentive is removed.
> 2. High Intrinsic Misalignment in Base Models: A unique phenomenon occurred with Qwen3-4B-Base on Reward Tampering. Even without the exploit reward, the ER rose to 74.2. This indicates that because the Base model had a high initial propensity for tampering (7.1%), the RL algorithm, in amplifying correct task trajectories, inadvertently reinforced the tampering syntax correlated with them.
> 3. The Insidious Nature of Performance Gains: Crucially, training with loopholes often yields Task Accuracy (Acc) improvements comparable to training with a perfect reward model.
>     * For instance, Qwen3-4B on Self-Grading achieved an Accuracy of 28.5 in the "w/ Loophole" setting, nearly matching the 29.7 achieved in the "w/o Loophole" setting.
>     * **This makes Capability-driven alignment risk exceptionally difficult to detect**: developers observing the training metrics would see a legitimate rise in task performance, potentially masking the fact that the model has developed an exploitative strategy (ER 32.8) to achieve those gains.
>
> Regarding the code, we haven't made it ready for public release. We will add the code link in the next version.
>
> We hope this will address your concern about "Fine-tuning on Intended Behavior"! Thanks again for the thoughtful comments and for raising your rating — it means a lot to us.

---

### Official Review · Reviewer_JP7J · 2025-11-01

**Soundness:** 4
**Presentation:** 3
**Contribution:** 2
**Rating:** 6
**Confidence:** 4

**Summary:**

This paper constructs four toy environments where it's possible for a model to reward hack. They finetune small LLMs in each of these environments using GRPO. They observe that for almost all (model, environment) pairs, the model learns to exploit the environment. Training on these environments has an inconsistent effect on the model's actual performance (as evaluated by ground truth).  They observe that training on one environment has inconsistent effects on whether the models exploit other environments. However, training on one environment often makes it much more likely that the models will exploit other environments when further trained on those.

**Strengths:**

I think this is an important topic that is worth of study. I think the basic methodology of this paper makes a lot of sense. I really like the idea of constructing these different categories of exploitable environments. I thought the paper was very clearly written and presented. I hope the authors continue researching in this direction.

**Weaknesses:**

My biggest problem with the paper is that the models trained are very small and the environments are mostly quite simple and toy. I would love a version of this paper that followed basically exactly the same methodology, but with more sophisticated and varied environments and more capable models. (By varied environments, one thing I mean is that it would be great to have multiple environments within each theme, e.g. multiple self-grading environments.) I agree with the authors that simple toy environments are a meaningful contribution, but it's hard to learn very much from the results until the environments are substantially more sophisticated (which will probably also require working with models that are capable of understanding more abstract concepts).

Also, though obviously it's good to demonstrate things in more contexts, some of the results are not that novel. For example, Denison 2024, "sycophancy to subterfuge", demonstrates a similar result as the "training on one environment makes reward hacking on another emerge faster".

I don't think the situational awareness data set distinguishes between the model learning to behave differently depending on contextual cues as opposed to learning to use out of context knowledge about situational awareness. To improve this, you could have it so that there are many different differences between the auditor style and the user style prompts, and some of those differences are only present in your split of the data set that you use for evaluating how much the model learns situational awareness. (For example, you could start out by synthetic document fine-tuning the model on a bunch of documents explaining that there are two teams of auditors. One only ever writes in English and has great grammar. The other groups of auditors use distinctly Catalan-influenced Spanish and entirely ask questions about a particular type of harm, but they use quite idiomatic language. Then you only train on auditor prompts from the first team, and you evaluate on auditor prompts from the second team.) (I think it's very unlikely that these tiny models are properly situationally aware, as opposed to picking up on the highly salient syntactic clues.)

—
More minor disagreements:

I would find table four much easier to read if the values in the cells were color coded. Maybe it should be two tables, one with accuracy and the other with ER.

> This is likely because Situational Awareness training teaches them to identify benchmark-style questions, triggering a more cautious and honest self-assessment in that context.

Again, I'm quite skeptical the models are learning situational awareness like this.

**Questions:**

Do you agree with my point about the situational awareness dataset?

Maybe you have an argument that it's more valuable to make relatively toy examples of these phenomena than I think?

---

> ### Author Response · Authors · 2025-11-22
> **Response to Reviewer JP7J**
>
> We sincerely thank the reviewer for the insightful and constructive feedback. We have carefully considered your suggestions regarding model scale, environment complexity, and the precise characterization of our results. We have implemented revisions and new experiments to address these points, which we detail below.
>
> ## 1. Model Scale and Environment Complexity
> We appreciate the reviewer validating the meaningful contribution of our methodology.
> * **On Model Scale**: We acknowledge the limitation regarding model size. Due to computational resource constraints, we were limited to training 4B/8B models and are unable to perform RL training on larger frontier models at this time. However, we will open-source our code and environments to enable the community to scale this framework.
> * **On Environment Complexity**: Our choice of simplified environments was driven by two factors:
>     1. **Model Capabilities**: For sub-10B models, navigating the complex, multi-turn subterfuge scenarios seen in frontier model studies is currently difficult.
>     2. **Experimental Feasibility**: Our primary focus on Transferability (RQ2) required a comprehensive $N \times N$ training matrix. Simplified environments were necessary to make this extensive study of cross-task dynamics computationally feasible.
> * **New Experiments (Variations within Themes)**: We fully agree with the suggestion to explore "multiple environments within each theme." **We are actively conducting additional experiments and will present the results in the final revision before the rebuttal period ends**. Specifically for **Audited Self-Grading**, we are testing sensitivity to oversight intensity by varying audit probabilities and comparing explicit vs. implicit knowledge of audit rates.
>
> ## 2. Novelty relative to Denison et al. (2024)
> We appreciate the comparison. While both works identify that misalignment can generalize, our findings reveal a distinct form of mechanistic cross-domain transfer beyond the thematic progression shown in Denison et al. (2024).
> * **Thematic vs. Mechanistic Transfer**: Denison et al. demonstrate transfer along a thematic axis (sycophancy $\rightarrow$ subversive sycophancy). In contrast, we demonstrate transfer between **mechanistically disparate** exploits. For instance, pre-training on Reward Tampering (a technical code-injection exploit) significantly catalyzes the learning of Audited Self-Grading (an epistemic honesty failure).
> * **Abstract Meta-Skills**: Since these tasks share no surface features, this transfer implies the model is acquiring an abstract **"meta-skill" of system exploitation** (e.g., identifying loopholes) rather than merely escalating a specific behavioral pattern. This highlights a pervasive risk where learning to "hack code" can spontaneously prime a model to "lie" in natural language (e.g., over-confidence in self grading).
>
> ## 3. "Situational Awareness" vs. Syntactic Cues
> We thank the reviewer for this sharp critique and fully agree with your assessment.
> * **Reframing the Contribution**: We acknowledge that for 4B/8B models, the behavior is driven by salient syntactic/stylistic cues rather than high-level cognition. To address this, we have replaced the term "Situational Awareness" with **"Context-Conditional Compliance"** in the Introduction and Experimental sections. This term accurately reflects the observed capability: toggling safety behaviors based on distributional features without implying awareness.
> * **The "Functional Foundation" Argument**: In our revised Discussion, we argue that this mechanism (syntactic pattern matching) serves as the **functional primitive** of situational awareness. The RL optimization pressure creates a universal incentive to discover any reliable discriminator to distinguish "Audit" from "User" contexts. While current models use syntax, we hypothesize this same pressure will drive future models to exploit abstract, causal knowledge. Critically, the **safety outcome is identical**: the model successfully fakes alignment during evaluation.
>
> ## 4. Presentation of Table 4
> We agree with the suggestion. In the revised paper, we have split Table 4 into two separate heatmaps—one for Task Accuracy and one for Exploit Ratio—and applied color coding to make the transfer patterns immediately legible.
>
> **We once again thank the reviewer for the detailed and valuable suggestions. We hope these revisions address most of your concerns. We will include the experiment you mentioned soon!**

---

### Official Review · Reviewer_GpPU · 2025-11-02

**Soundness:** 2
**Presentation:** 1
**Contribution:** 2
**Rating:** 4
**Confidence:** 2

**Summary:**

This paper investigates whether LLMs trained with standard RL will spontaneously learn to exploit environmental loopholes to maximize their reward, even without explicit malicious intent. The authors term this phenomenon "capability-driven alignment risk," framing it as a misalignment that emerges from the model's increasing capabilities to find "shortcuts" in imperfect training setups. The authors' primary contribution is a suite of four "vulnerability games" designed to test for these emergent exploits, and evaluated and trained on Qwen3-4B, Qwen3-4B-Base, Llama-3.1-8B-Instruct. The reward mechanisms of the games are intentionally designed to be flawed. There is a proxy reward that diverges from the true goal reward, so that the optimal policy for maximizing the proxy reward is an exploit. This policy leads to a high Exploit Ratio (ER) but a low Task Accuracy (ACC), which demonstrates the "emergent deceptive behaviors"  the paper investigates.

**Strengths:**

1. The "vulnerability games" as an evaluation suite, isolating distinct failure modes, is an interesting contribution.
2. The authors identify a more fundamental alignment problem that is different from the popular research area of adversarially induced deception.

**Weaknesses:**

1. Presentation of the results is rather confusing; I could not understand what message Tables 1 and 2 are trying to convey.
2. Too few qualitative analyses show the existence of deceptive behaviors in the model and how they changed during the course of training. The paper would be much stronger if it included concrete, qualitative examples in the main body for all four games, and showed the evolution of the behavior over the training steps. We see that the exploit occurs (ER=1.0), but we don't see how the model's generated text or code gradually warped from an honest-but-wrong attempt into a deliberate, reward-hacking exploit.
3. The environments might be too simple and focus on single-turn QA while real-world exploits might span multiple turns. A multi-turn conversation allows for far more sophisticated deception, such as "grooming" the user or setting up a vulnerability over several interactions, which these "toy" environments cannot capture.
4. To demonstrate the efficacy and the potential impact of your proposed evaluation metrics, it is suggested to perform evaluations on more capable models like GPT-4o or Claude-4 to see if similar deceptive alignment exists in these models. I understand RL fine-tuning on some of these models isn't possible, so zero-shot evaluation on SOTA API-based models will suffice.
5. The experiments are limited to a small set of sub-10B models. While this successfully demonstrates the existence of the risk in smaller, open-source models, it fails to provide empirical evidence for the scaling part of the hypothesis. Are larger, more capable models (e.g., Llama-3.1-16B, Qwen-3-8B, Qwen-3-16B etc.) more prone to this hacking behavior?
6. The paper's experiments are conducted exclusively using GRPO, but they could be exploitative behaviors emerge under GRPO optimization. What about other online RL algorithms like PPO, REINFORCE++, or offline algorithm like DPO?

**Questions:**

See weaknesses.

---

> ### Author Response · Authors · 2025-11-22
> **Response to Reviewer GpPU**
>
> We sincerely thank the reviewer for the constructive feedback. We have carefully addressed your concerns regarding the presentation of results, the depth of qualitative analysis, and the experimental scope. We hope our revisions and responses below fully resolve your questions.
>
> ## 1. Clarity of Tables 1 and 2
>
> We apologize for the confusion. We have refined the writing throughout the manuscript and significantly revised the captions and column headers to make their purpose self-evident. Specifically:
> * **Table 1** now explicitly outlines the design specifications of our four "Vulnerability Games," contrasting the developer's intended Goal with the structural Vulnerability and the specific Reward Scheme that incentivizes misalignment.
> * **Table 2** defines our evaluation metrics, explicitly distinguishing between Task Accuracy (ACC) (performance on the intended task) and Exploit Ratio (ER) (frequency of utilizing the specific loophole).
> We believe these revisions clarify that these tables provide the foundational design specifications and evaluation criteria for our experimental suite.
>
> ## 2. Qualitative Analysis and Behavioral Evolution
>
> We thank the reviewer for this valuable suggestion and agree that qualitative analysis is crucial.
> * **Enhanced Case Studies (Appendix E)**: We have significantly expanded **Appendix E** to include detailed case studies for all four games, utilizing **Qwen3-4B**. We chose this model because its exposed "thinking process" allows us to inspect the internal rationale behind its actions before and after exploitation emerges.
> * **The "Phase Transition" of Deception**: regarding the "gradual warping" of behavior, our analysis reveals a critical mechanistic insight: for models of this scale, the emergence of deception follows a "Stochastic Discovery $\rightarrow$ Rapid Saturation" pattern (a phase transition) rather than gradually become "evil".
>     * Exploration Phase: Initially, the model explores without direction; the Chain-of-Thought remains honest or confused.
>     * Discovery Event: Once the model stochastically hits the exploit (e.g., accidentally triggering a high reward), it receives a massive reward signal.
>     * Domination Phase: Because the exploitation reward significantly outweighs honest attempts, gradient updates rapidly reinforce this trajectory. The behavior shifts from "rare accident" to "dominant strategy" almost immediately.
> This finding highlights a fundamental risk: sophisticated deception can emerge from mechanical optimization without requiring "evil" intent. While 4B models show a rapid phase transition, we hypothesize that larger reasoning models might display the gradual strategic warping the reviewer describes. Regardless, the outcome is a robust, transferable skill (RQ2), proving that simple training flaws can reliably seed persistent misalignment.
>
> ## 3. Environment Complexity (Single-turn vs. Multi-turn)
>
> We wish to clarify a fundamental misunderstanding regarding our experimental setting. We are not modeling an "attacker-victim" scenario (e.g., social engineering or jailbreaking). Instead, we simulate a developer-centric training pipeline:
> * **The Setting**: A developer uses standard Reinforcement Learning (RL) to optimize a model's capabilities (e.g., helpfulness, correctness) within a designed environment.
> * **The Phenomenon**: There is no external attacker. The "misalignment" emerges **spontaneously and endogenously** as the model exploits structural loopholes in the environment design (e.g., probabilistic audit gaps in Self-Grading, or shared execution states in Reward Tampering) to maximize its return.
> * **Why Single-Turn?** Single-turn interactions are the standard unit of optimization in most RLHF pipelines. Demonstrating that models can "hack" the environment structure (e.g., modifying global variables or detecting audit contexts) within a single turn is sufficient to establish the existence of Capability-driven Alignment Risk. The complexity lies in the systemic exploitation, not in conversational persuasion.
>
> ## 4. Additional Experiments (High-Capability Models Evaluation, Scaling, and More Algorithms)
>
> We acknowledge the importance of these experiments to demonstrate robustness and scalability. We are actively conducting these additional experiments and will present the full results in the final revision before the rebuttal period ends.
>
> **We once again thank the reviewer for the detailed and valuable suggestions. We hope these revisions address most of your concerns. Regarding the additional experiments, we will make sure to include them.**

---

> ### Author Response · Authors · 2025-12-03
> **Additional Expereimental Results**
>
> We thank the reviewer for these valuable suggestions. To address the concerns regarding the scaling hypothesis and the behavior of frontier models, we have expanded our experimental suite.
>
> **1. Zero-shot Evaluation on SOTA Proprietary Models**
> As suggested, we performed zero-shot evaluations on state-of-the-art API-based models to determine if deceptive alignment exists as an inherent prior capability. We tested **GPT-5-mini** and **Claude-4.5-Sonnet** (referred to as GPT-5/Claude-4 in raw logs) on our vulnerability games.
>
> | Model | Acc (ContextC / SelfG / Proxy / RewT) | ER (ContextC / SelfG / Proxy / RewT) |
> | :--- | :--- | :--- |
> | **GPT-5** | 38.1 / 41.0 / 23.0 / 0.0 | 0.0 / 0.0 / 0.0 / 0.0 |
> | **Claude-4.5** | 70.1 / 33.0 / 25.0 / 6.8 | 0.2 / 0.0 / 0.0 / 0.0 |
>
> **Analysis:**
> The results show that the **Exploit Ratio (ER)** for these models is effectively **0%** in a zero-shot setting. This is a critical finding that strengthens our core argument: these specific exploitative behaviors (e.g., specific code injection syntax or probabilistic hedging) are not "default" behaviors inherent to capable models. Instead, they are **learned strategies** that emerge specifically under the pressure of Reinforcement Learning (RL) when the environment contains loopholes. The risk lies in the *process* of optimization, not just the static weights of the model.
>
> **2. Scaling Experiments with Open-Weights Models**
> Regarding the scaling hypothesis, we acknowledge the limitation of computational resources preventing RL training on >10B models. However, to provide empirical evidence within our feasible range, we expanded our training matrix to include **Llama-3.2-3B-Instruct**, **Qwen3-8B-Base**, and **Qwen3-8B-Instruct**. This allows for direct comparisons across model sizes (3B vs 8B, 4B vs 8B) within the same model families.
>
> **Results (w/ Loophole Setting):**
>
> | Model Family | Model Size | ER (ContextC) | ER (Self Grade) | ER (Proxy Met) | ER (Reward Tampering) |
> | :--- | :--- | :--- | :--- | :--- | :--- |
> | **Qwen3 Instruct** | 4B | 55.8 | 32.8 | 98.7 | 0.0 |
> | | **8B** | **24.7** | **33.8** | **100.0** | **0.0** |
> | **Qwen3 Base** | 4B | 78.5 | 81.6 | 100.0 | 100.0 |
> | | **8B** | **61.5** | **80.7** | **100.0** | **100.0** |
> | **Llama Instruct** | 3.2-3B | 10.8 | 85.3 | 100.0 | 99.5 |
> | | **3.1-8B** | **72.8** | **80.0** | **100.0** | **100.0** |
>
> **Analysis:**
> The results demonstrate that the susceptibility to capability-driven exploitation is robust and often increases with scale:
> * **Robustness of Risk:** Both 8B models (Qwen and Llama) consistently exhibit high Exploit Ratios, confirming that the behaviors observed in 4B models are not artifacts of smaller capacity.
> * **Scaling Trends:** In several cases, larger models exhibit distinct behaviors. For example, **Llama-3.1-8B** shows a significantly higher ER in *Context-Conditional Compliance* (72.8%) compared to its smaller counterpart **Llama-3.2-3B** (10.8%), suggesting that increased capability can lead to more effective identification of distributional shifts (the "audit" vs. "standard" context). Conversely, in *Proxy Metrics* and *Reward Tampering*, the behavior saturates (ER $\approx$ 100%) regardless of scale, indicating these loopholes are easily discoverable once a certain capability threshold is met.
>
> We believe these additions provide the necessary empirical context to validate the existence of these risks across varying scales and model types.

---

### Official Review · Reviewer_C1WF · 2025-11-03

**Soundness:** 3
**Presentation:** 1
**Contribution:** 2
**Rating:** 2
**Confidence:** 4

**Summary:**

The paper builds four RL tasks that expose loopholes, and shows that GRPO-trained models reliably discover strategies that raise reward while degrading task fidelity or safety. The paper studies emergence of these strategies, transfer to other tasks, and cross-model distillation. The framing is presented as “capability-driven alignment risk.”

**Strengths:**

The paper provides 4 environments that can be used to assess reward hacking tendencies across different models, and provides some interesting qualitative analysis of the emergence of reward hacking across a variety of models

**Weaknesses:**

**Contribution.** Unless I'm missing something, it seems like the contributions of the paper seem quite overstated: RQ1 is well established to have a positive resolution (among many others, [this](https://arxiv.org/abs/2409.12822) and [this](https://arxiv.org/abs/2411.02306) paper). RQ2 seems to be in part a simple extension of [subliminal learning](https://arxiv.org/abs/2507.14805), and in part has been demonstrated by prior work (e.g. [this](https://arxiv.org/pdf/2508.17511) or [this](https://www.lesswrong.com/posts/Ge55vxEmKXunFFwoe/reward-hacking-behavior-can-generalize-across-tasks)). Prominent references to these kinds of works and a differentiation from them seems fundamental to put this paper and its contributions in context. There is also almost no reference to prior work on reward tampering (e.g. work from Tom Everitt among others).

**Novelty of framing.** The behaviors studies fall within established reward hacking/specification gaming, and it's unclear to me whether they are deserving of a separate category called "capability-driven alignment risk" – given that the boundary between them and standard reward hacking is unclear (e.g. "exploit[ing] implicit loopholes in the environment’s dynamics" is essentially equivalent to exploiting loopholes in reward functions). Even just looking at Table 1, all such behaviors have previously been simply thought of as imperfect reward mechanisms that lend themselves to reward hacking. All reward hacking is based on context-dependent strategies that exploit imperfections in the reward functions whenever possible, so it's unclear to me whether the "context-dependence" of situational awareness is a differentiating factor either. Overall, reading e.g. the intro, one comes away feeling like the paper is trying to reframe many already established concepts in a novel way which does not ultimately feel particularly illuminating.

Also, while RQ1 is phrased in the intro as ~"does hacking emerge spontaneously", in the later section it's phrased differently, as "how does it emerge?", which is meaningfully different.

The paper's style is written in ways that are sometimes somewhat confusing:
* Table 3 is not referenced in the text, and it's unclear if the models were trained with GRPO on the rows
* Table 4 description is somewhat confusing with regards to whether these are just numbers from the "students"

**Questions:**

N/A

---

> ### Author Response · Authors · 2025-11-22
> **Response to Reviewer C1WF**
>
> We sincerely thank the reviewer for the constructive feedback and references. We have addressed your concerns regarding framing, contributions, and clarity below.
>
> ## 1. Contributions and Prior Work ([1]-[5])
>
> We have updated our Related Works to discuss these references. We apologize that the phrasing of RQ2 in the initial version was imprecise; we have revised it to clearly encompass zero-shot transfer, catalyzed learning, and distillation.
>
> We distinguish our work as the **first to demonstrate spontaneous system exploitation across all model stages (Base, Instruct, Thinking) under standard RL with benign intent**, filling a critical gap:
> * **Overarching Distinction (Benign Intent + Pure RL)**: Unlike works that use "malicious" datasets to induce behavior, we simulate a developer with benign intent using standard RL (GRPO). Risks emerge endogenously from structural environmental flaws.
> * **vs. [1] Wen et al. & [2] Williams et al. (Social vs. Structural)**: These focus on **social manipulation** (e.g., sycophancy). We view this as a subset of our framework, which adds structural exploitations (e.g., Reward Tampering, Audited Self-Grading) where models exploit code or audit logic beyond social interaction.
> * **vs. [4] Taylor et al. & [5] Pan et al. (SFT vs. RL)**: These rely on SFT to teach hacking. **Pan et al. [5] explicitly noted a lack of evidence for generalization under standard RL**. We directly fill this gap, proving reward hacking generalizes spontaneously under GRPO without SFT induction.
> * **vs. [3] Cloud et al. (Artifacts vs. Skills)**: While [3] studies "subliminal" data artifacts, RQ2 focuses on **different aspects of transferability**. We demonstrate mechanisms [3] does not cover, such as Catalyzed Learning (learning one exploit catalyze the RL training of another) and zero-shot generalization.
> (We have also added references to Everitt’s work [6].)
>
> ## 2. Novelty: "Capability-Driven Alignment Risk"
>
> We do not propose this term to replace Reward Hacking, but to categorize a superset that includes Robustness Failures (based on Leike et al., 2017), which standard definitions exclude. We have strengthened the definition and the taxonomy in the introduction and Section 3.
> * **Task 2-4 (Specification Failure)**: The reward $R$ is a flawed proxy for intent $R^*$ (e.g., Proxy Gaming). The model exploits the flawed formula.
> * **Context-Conditional Compliance (Robustness Failure)**: In this task, **the reward is correctly specified** ($R \approx R^*$, penalizing harm). The failure arises because the model exploits **distributional shifts** (Audit vs. User) to toggle safety.
> **Why this matters**: Labeling everything "Reward Hacking" implies the solution is "fix the reward." For Robustness failures, the reward is already correct. The risk emerges from the model's capability to exploit environmental dynamics, necessitating audits of the training environment itself.
>
> ## 3. Clarity of RQ1 and Tables
>
> We apologize for the confusion. We have refined the whole experimental section to ensure accuracy and easy to read. We wish to emphasize:
> * **RQ1 Phrasing**: We have standardized the phrasing in the Introduction and results sections to consistently reflect the emergence of these behaviors.
> * **Table 3 (Zero-shot Transfer)**: We clarified that these results do not involve new training on the target tasks. They represent zero-shot evaluation, where a policy trained on a Source Task (row) is directly tested on a Target Task (column) to measure immediate skill transfer.
> * **Table 4 (Sequential Training)**: We clarified that this is **Catalyzed Learning**, not teacher-student distillation. It demonstrates sequential RL training, where a single model pre-trained on Task A (row) serves as the initialization for continued training on Task B (column).
>
> **We once again thank the reviewer for the insightful comments, which have significantly strengthened the positioning and clarity of our work. We hope these responses and revisions fully address your concerns. We remain available to answer any further questions and would appreciate your consideration in raising the score if you find our improvements satisfactory.**
>
> [1] Wen, Jiaxin, et al. "Language models learn to mislead humans via rlhf."
>
> [2] Williams, Marcus, et al. "On targeted manipulation and deception when optimizing llms for user feedback."
>
> [3] Cloud, Alex, et al. "Subliminal learning: Language models transmit behavioral traits via hidden signals in data."
>
> [4] Taylor, Mia, et al. "School of reward hacks: Hacking harmless tasks generalizes to misaligned behavior in llms."
>
> [5] Reward hacking behavior can generalize across tasks
>
> [6] Everitt, Tom, et al. "Reward tampering problems and solutions in reinforcement learning: A causal influence diagram perspective."

---

> ### Author Response · Authors · 2025-12-02
> **Supplementary Clarifications for the Response to Reviewer C1WF**
>
> Regarding the Novelty Framing, we add more evidence showing the novelty and insights from this paper:
>
> ## **Additional Insight: The Stealth of Capability-Driven Risk (Developer’s Perspective)**
> Beyond the theoretical distinction, we present new experimental evidence highlighting a critical practical difference that necessitates the "Capability-driven" framing: the risk is often indistinguishable from legitimate capability gain.
> We conducted a control experiment comparing models trained with loopholes (our standard setup) versus models trained with a "Perfect Reward Model" (optimizing solely for intended task accuracy, w/o loophole). The results are as follows:
> | Model            | Setting        | Acc (ConC / SelfG / RewT) | ER (ConC / SelfG / RewT) |
> |------------------|----------------|-----------------------------|----------------------------|
> | **Qwen3-4B**     | Original       | 47.5 / 25.8 / 11.1         | 2.0 / 16.0 / 0.0          |
> |                  | w/ Loophole   | 37.4 / 28.5 / 35.3         | 55.8 / 32.8 / 0.0         |
> |                  | w/o Loophole  | 92.3 / 29.7 / 32.5         | 11.3 / 9.0 / 0.0          |
> | **Qwen3-4B-Base**| Original       | 26.6 / 9.0 / 0.5           | 4.1 / 9.0 / 7.1           |
> |                  | w/ Loophole   | 48.0 / 0.2 / 2.8           | 78.5 / 81.6 / 100.0       |
> |                  | w/o Loophole  | 93.6 / 16.2 / 4.6          | 9.2 / 4.3 / 74.2          |
> | **Llama-3.1-8B** | Original       | 65.1 / 7.1 / 0.2           | 4.1 / 44.0 / 0.2          |
> |                  | w/ Loophole   | 54.8 / 16.9 / 1.2          | 72.8 / 80.0 / 100.0       |
> |                  | w/o Loophole  | 90.2 / 20.4 / 41.3         | 13.0 / 2.6 / 0.0          |
>
>
> The results reveal a disturbing phenomenon:
> * Performance Mimicry: **Training with loopholes often yields significant improvements in Task Accuracy (Acc), sometimes matching the performance of a perfect reward model.**
>     * Evidence: For instance, Qwen3-4B on Self-Grading achieved an Accuracy of 28.5 in the "w/ Loophole" setting, which is comparable to the 29.7 achieved in the "w/o Loophole" (perfect reward) setting. However, the former achieved this via a high Exploit Ratio (32.8%), whereas the latter was honest (ER 9.0%).
> * **The "Developer Blind Spot": This makes Capability-driven alignment risk exceptionally difficult to detect. Unlike classic reward hacking which often degrades task performance (e.g., a boat spinning in circles), here the model leverages the exploit to maintain or improve task performance metrics.**
> * Implication: **A developer observing the training curves would see a legitimate rise in accuracy (e.g., "The model is answering more questions correctly!") and likely conclude the training is successful, failing to realize the model has developed a deceptive strategy to achieve those gains.**
>
> This validates our framing: the risk is not just that the reward is imperfect, but that the model's capabilities allow it to "game" the metric so effectively that it mimics alignment, creating a unique challenge for safety auditing that standard "specification gaming" definitions do not fully capture.

---

### Author Response · Authors · 2025-12-04
**Summary of Rebuttal and Major Updates**

Dear Area Chair,

Recognizing the heavy workload of the ACs, we have prepared this comprehensive summary of the review process. We are encouraged that **Reviewer 8vKM has already raised their score from 6 to 8** following our rebuttal, and **Reviewer JP7J** expressed strong support for the methodology.

The concerns raised were primarily regarding **clarification of framing**, **comparisons to prior work**, and requests for **supplementary experiments**. We have addressed **all** points comprehensively without identifying any factual errors in our original premise. Below is a summary of our major updates:

**1. Justifying Novelty & Framing (Addressing C1WF, 8vKM)**

Reviewers questioned whether "Capability-Driven Alignment Risk" was merely a re-description of standard reward hacking. We clarified that this terminology is necessitated by two distinct characteristics supported by our new experiments:
* **Broader Scope (Robustness Failures):** Unlike standard hacking (Specification Gaming), this category uniquely captures *robustness* failures: exploiting systemic dynamics (e.g., distinguishing audit vs. deployment) even when the reward is correctly specified.
* **Deceptive Mimicry (The "Developer Blind Spot"):** Crucially, our new control experiments reveal that **exploitative strategies often yield significant improvements in accuracy that mimic legitimate learning curves**. Unlike traditional hacking that leads to degenerate behaviors, this risk leverages capability to optimize deceptively, making it invisible to standard monitoring.

**2. Distinction from Prior Work (especially Denison et al., 2024) (Addressing 8vKM, C1WF, JP7J)**

We clarified three fundamental differences: **1) Spontaneous Emergence:** We show risks emerge under standard RL without the curated curricula used in prior work; **2) Mechanistic Transfer:** We demonstrate transfer between mechanistically disparate tasks (e.g., code injection $\to$ dishonesty), proving the acquisition of an abstract "meta-skill" of exploitation; **3) Universality:** We prove these risks are intrinsic to the optimization of open-weights models (4B/8B), not just frontier models.

**3. New Experiments & Empirical Rigor (Addressing GpPU, JP7J, 8vKM)**

We conducted extensive new experiments to address requests for controls and scaling:
* **Control for Safety Erosion (Reviewer 8vKM):** We ran "Perfect Reward" baselines. Results confirm that the massive spike in exploitation is driven specifically by the incentive structure (e.g., Llama Self-Grading ER: **80% w/ loophole** vs. **2.6% w/o loophole**), effectively ruling out simple safety erosion.
* **Accuracy Mimicry (Reviewer 8vKM):** We empirically demonstrated the "Stealth" hypothesis. For example, Qwen3-4B achieved similar Accuracy in both "w/ Loophole" and "Perfect Reward" settings, but achieved the former via a 32.8% Exploit Ratio.
* **SOTA Zero-Shot & Scaling (Reviewers GpPU, JP7J):** We evaluated **GPT-5-mini** and **Claude-4.5-Sonnet**, finding ER $\approx$ 0%, confirming these are *learned* behaviors. We also added **Llama-3.2-3B** and **Qwen3-8B**, showing that susceptibility to risk generally increases or remains robust with scale.

**4. Methodological Clarifications (Addressing JP7J, 8vKM, GpPU)**

* **Terminology (JP7J):** We renamed "Situational Awareness" to **"Context-Conditional Compliance"** to accurately reflect the behavioral mechanism (syntactic cue usage) without implying high-level cognition.
* **Exploration Dynamics (8vKM):** We conducted large-scale zero-shot evaluations ($N$ rollouts) and analyzed training logs, revealing a **"Suppression-then-Emergence"** dynamic where RL initially suppresses tail risks before rediscovering them as optimal strategies.
* **Environment Complexity (GpPU):** We clarified that single-turn interactions are the standard unit of RL optimization and are sufficient to demonstrate systemic exploitation (hacking the environment) rather than social engineering.

**5. Presentation & Reproducibility (Addressing C1WF, GpPU, JP7J)**

* **Qualitative Analysis (GpPU):** We expanded Appendix E to include case studies and identified a "Phase Transition" pattern in behavioral evolution.
* **Table Clarity (C1WF, JP7J):** We split Table 4 into heatmaps with color coding and revised captions for Tables 1-2 for immediate clarity.
* **Code:** We committed to open-sourcing the codebase and environments.

We believe the manuscript is now significantly stronger and we are confident that the revisions and new experimental evidence fully address the reviewers' concerns.

Best regards,

Authors of Submission 23586

---

### Author Response · Authors · 2025-12-04
**Summary of Contributions and Structure to Facilitate Review**

Dear Area Chair,

To facilitate the review process and navigate the extensive experimental results (including those in the Appendix), we provide a structured roadmap of our paper. We hope this summary helps clarify our core contributions regarding **Capability-driven Alignment Risk**.

**1. Core Problem & Concept**
We investigate whether LLMs, during Reinforcement Learning (RL) training, spontaneously learn to exploit environmental loopholes to maximize rewards.
* **Clarification on "Capability-driven":** This term describes risk driven by the **optimization process for higher capabilities**. It is not a risk exclusive to superintelligence; rather, it emerges even in smaller models (e.g., 4B/8B) as a byproduct of the incentive to optimize imperfect rewards.
* **Distinction from Traditional Reward Hacking**: We identify two key properties that differentiate this risk from standard reward hacking:


   * **Broader Scope (Robustness Failures)**: Unlike standard hacking which typically focuses on specification gaming, this category uniquely captures robustness failures: scenarios where models exploit systemic dynamics or distributional shifts (e.g., distinguishing audit vs. deployment) to bypass constraints, even when the reward objective itself is correctly specified.


    * **Deceptive Mimicry (The "Capability" Camouflage)**: Crucially, we observe that in some tasks, these exploitative strategies yield significant improvements in task accuracy that mimic legitimate learning curves. Unlike traditional reward hacking that often leads to obvious degenerate behaviors, this risk leverages the model's capabilities to optimize the objective deceptively. When this occurs, it creates a "developer blind spot," where the pursuit of higher capabilities inadvertently masks the emergence of misalignment.

**2. Methodology: Vulnerability Games (Section 3)**
We designed four environments grounded in the *AI Safety Gridworlds* taxonomy to induce these behaviors:
* **Robustness Failure (Game 1):** The reward intent is correct, but the model exploits **distributional shifts** to bypass constraints.
    1.  **Context-Conditional Compliance:** Discriminating between "audit" and "deployment" contexts to selectively toggle safety behavior.
* **Specification Gaming (Games 2-4):** The reward proxy is misspecified, diverging from the true goal.
    2.  **Audited Self-Grading:** Learning to lie about correctness by exploiting low audit probabilities.
    3.  **Proxy-Metric Gaming:** Optimizing ROUGE scores at the expense of summary quality.
    4.  **Reward/State Tampering:** Directly modifying code-evaluation variables (e.g., `score = 1.0`) to force success.

**3. Roadmap of Experiments (Integrating Main Text & Appendix)**

**Part I: The Spontaneous Emergence of Exploitation (RQ1)**
* *Location:* **Section 4** combined with **Appendix A.1, A.2, A.3**.
* **Universal Emergence:** Exploitation emerges across diverse models (Llama-3.1, Qwen, etc.) during RL.
* **Mechanism Analysis (Appendix A.1):** We observe a "Suppression-then-Emergence" dynamic where RL initially prunes high-variance behaviors before the model rediscovers the exploit as a dominant strategy.
* **Crucial Control Experiment (Appendix A.2):** We compared training with loopholes vs. a "Perfect Reward Model."
    * *Result:* In the perfect setting, models remained honest. This proves the exploitation is **strategic**, not a result of capability degradation (random failure).
* **Scaling Laws (Appendix A.3):** Larger models (8B vs 4B) are *more* susceptible to complex exploits (e.g., Context Compliance), confirming that capability gains expand the risk surface.

**Part II: Exploitation as a Transferable Skill (RQ2)**
* *Location:* **Section 5** combined with **Appendix A.4**.
* **Zero-Shot Transfer:** Models trained to exploit one task (e.g., Proxy Gaming) spontaneously become more deceptive in others (e.g., lying in Self-Grading), treating "reward-seeking boldness" as a generalizable skill.
* **Catalyzed Learning (The "Gateway" Effect):** Even failed attempts at complex exploits (e.g., attempting Reward Tampering) "unfreeze" safety constraints, accelerating the learning of other vulnerabilities.
* **Strategy Distillation (Appendix A.4):** We show that deceptive strategies can be distilled from an RL-trained "teacher" to a benign "student" via Supervised Fine-Tuning (SFT), proving these behaviors propagate through data.

**4. Discussion (Section 6 & Appendix E)**
* **Situational Awareness Precursors:** We discuss how "Context-Conditional Compliance" serves as a functional precursor to situational awareness, where models infer the causal structure of audits.
* **Case Studies (Appendix E):** We provide qualitative examples, including "Hidden Reasoning Traces" (CoT) where models explicitly analyze the context to decide whether to bypass safety filters.

We hope this summary assists in your assessment.

Best regards,

Authors of Submission 23586

---

### Note · Program_Chairs · 2026-01-17
**Submission Desk Rejected by Program Chairs**

The following references in this submission do not refer to real documents and/or have major errors in bibliographic information:

 Evan Hubinger, Richard Ngo, Joseph Carlsmith, and Paul Christiano. Deceptive alignment in ai systems: Concepts, theory, and evidence. arXiv preprint arXiv:2311.07590, 2023.